# Isolation and Characterization of *Paenibacillus polymyxa* B7 and Inhibition of *Aspergillus tubingensis* A1 by Its Antifungal Substances

**DOI:** 10.3390/ijms25042195

**Published:** 2024-02-12

**Authors:** Tianyuan Zhao, Jianing Ma, Meiwei Lin, Chen Gao, Yuhao Zhao, Xin Li, Weihong Sun

**Affiliations:** College of Agricultural Engineering, Jiangsu University, Zhenjiang 212013, China; zhaotianyuan0711@163.com (T.Z.); majianing7581@163.com (J.M.); 2212216002@stmail.ujs.edu.cn (M.L.); 2212216007@stmail.ujs.edu.cn (C.G.); zhaoyuhao55@163.com (Y.Z.); 2212316010@stmail.ujs.edu.cn (X.L.)

**Keywords:** *Aspergillus tubingensis*, *Paenibacillus polymyxa*, antifungal proteins, biological control

## Abstract

Screening of *Bacillus* with antagonistic effects on paddy mold pathogens to provide strain resources for biological control of mold in *Oryza sativa* L. screening of *Bacillus* isolates antagonistic towards *Aspergillus tubingensis* from rhizosphere soil of healthy paddy; classification and identification of antagonistic strains by biological characteristics and 16S rDNA sequence analysis; transcriptome sequencing after RNA extraction from Bacillus-treated *Aspergillus tubingensis*; and extraction of inhibitory crude proteins of *Bacillus* by ammonium sulfate precipitation; inhibitory crude protein and *Bacillus* spp. were treated separately for *A. tubingensis* and observed by scanning electron microscopy (SEM). An antagonistic strain of *Bacillus*, named B7, was identified as *Paenibacillus polymyxa* by 16S rDNA identification and phylogenetic evolutionary tree comparison analysis. Analysis of the transcriptome results showed that genes related to secondary metabolite biosynthesis such as antifungal protein were significantly downregulated. SEM results showed that the mycelium of *A. tubingensis* underwent severe rupture after treatment with *P. polymyxa* and antifungal proteins, respectively. In addition, the sporocarp changed less after treatment with *P. polymyxa*, and the sporangium stalks had obvious folds. *P. polymyxa* B7 has a good antagonistic effect against *A. tubingensis* and has potential for biocontrol applications of paddy mold pathogens.

## 1. Introduction

Paddy (*Oryza sativa* L.) is an important food crop, and according to the Food and Agriculture Organization of the United Nations (FAO), the annual worldwide grain loss due to mold is 3%. In 2020, China produced more paddy annually than any other country, accounting for 28% of global paddy production [1]. After paddy harvesting and drying, the microorganisms in the paddy increase with storage time. *Aspergillus* spp. fungi such as *Aspergillus niger* [2], *Aspergillus flavus* [3,4], and *Aspergillus ochraceus* [5] produce toxins that contaminate the paddy during the storage period, thus affecting food safety [6]. Finding effective methods to control the contamination of such *Aspergillus* fungi is essential to ensure food safety. Currently, the common method for controlling *Aspergillus* fungal contamination is mainly the use of synthetic chemical fungicides [7]. However, chemical fungicides may lead to an increase in fungal resistance, and the residual presence of fungicides may pose a threat to people’s health. Unlike such methods, biocontrol provides a safe and environmentally friendly method of inhibiting fungal growth.

Many microbial strains, such as yeasts [8,9], bacteria [10], and filamentous fungi [11], have been used to control mold contamination [12]. Among the microbial strains associated with biocontrol mechanisms, the broad-spectrum antagonistic *Bacillus* spp. is emerging as one of the most valuable genera. This is due to its good biological activity and high inhibitory activity against various pathogenic microorganisms, and *Paenibacillus* spores survive for a long period of time and are resistant to heat, drying, toxic chemicals, and radiation [13,14,15,16,17]. Strains reported to have good antifungal activity include *Bacillus subtilis* fmbJ [18], *Bacillus velezensis* E2 [19], *Paenibacillus polymyxa* 7F1 [20], *Bacillus atrophaeus* J-1 [21], and *Bacillus amyloliquefaciens* MG3 [22]. The biocontrol mechanisms of *Bacillus* mainly involve competition between *Bacillus* and pathogens for nutrients and ecological niches [23,24,25], the production of antimicrobial substances [26], interaction between *Bacillus* and pathogens [27,28], induction of the plant’s systemic resistance [29,30], and so on. One of the main antifungal mechanisms in *Bacillus* is producing compounds with antifungal effects, such as inhibitory proteins [31], inhibitory lipopeptides [19], and other secondary metabolites [32]. Paenibacillus-derived antimicrobials include polymyxins and fusaricides, which are non-ribosomal lipopeptides first isolated from strains of *P. polymyxa*. Other useful molecules include exo-polysaccharides (EPSs) and enzymes [17]. *P. polymyxa* SK1 exhibited strong antifungal activity against several airborne fungal pathogens due to its production of multiple antimicrobial peptides [33]. The *P. polymyxa* strain is capable of emitting bacterial volatile compounds, suppressing plant pathogens [34,35]. Dibutyl phthalate (DBP) and di (2-ethylhexyl) phthalate (DEHP), components of *P. polymyxa* strains L1-9, showed significant inhibitory effects on five phytopathogenic fungi [36].

Antimicrobial proteins are important metabolites produced by some strains of the genus *Bacillus*. Studies have shown that a variety of *Bacillus* strains can produce antifungal proteins: an antifungal protein produced by the *P. polymyxa* strain VLB16 has caused severe alterations in cell morphogenesis that gave rise to swollen and rounded hyphae [37]; the effective antimicrobial substances isolated from *Bacillus velezensis* MCCC 1A15695 provide a reference for the prevention and control of shrimp diseases [38]; PBT1, a new protein isolated from *Bacillus subtilis* XF-1, has been found to inhibit *P. brassicae* and the mycelial growth of other plant-pathogenic fungi [39]; and an antifungal protein isolated from *B. licheniformis* HS10 and *Bacillus cereus* has demonstrated antifungal activity [40,41].

In our experiments, we selected the Japonica rice variety Nan Jing 46, which has the outstanding advantage of excellent rice quality and has been rated as a high-quality flavor of Japonica rice. Nan Jing 46 combines the flavor of the maternal Japonica 14 and the soft rice characteristics of the paternal Kanto 194, and it is suitable for cultivation in the southern part of Jiangsu Province, China. The aim of this study was to isolate and identify antagonistic bacteria against *Aspergillus* and to characterize their antifungal activities and antifungal mechanisms, as well as to study the mechanism of action of their inhibitory substances against dominant molds in paddy. By understanding the antifungal molecular mechanisms of *Bacillus*, we can lay the foundation for the further production and application of antifungal proteins, etc., which will ensure food safety and reduce the contamination and loss of paddy.

## 2. Results

### 2.1. Dominant Molds

Screening counts revealed that the dominant molds of paddy were *Aspergillus* spp. fungi such as *A. tubingensis* (Figure 1A), *A. flavus* (Figure 1B), and *A. ochraceus* (Figure 1C). This coincides with the conclusions reached by Chai [42] and Qi [43] in their studies.

On PDA medium, the morphological characteristics of *A. tubingensis* are as follows: it is white at the beginning and black after 7 d of incubation (Figure 1A), with semi-tomentose edges. Morphological characteristics of *A. flavs*: it is yellow at the beginning and yellowish green after 7 d of incubation (Figure 1B), similarly with semi-tomentose edges, and spore production in the later stage. Morphological characteristics of *A. orchrue*: initially yellowish, and after 7 d of incubation (Figure 1C), it becomes dark yellow.

### 2.2. Identification of Antagonistic Bacteria Isolates and Inhibition of Mycelial Growth of A. tubingensis A1 with Scanning Electron Microscope (SEM) Evaluation

Seven strains with antagonistic effects were isolated from paddy rhizosphere soil. Strain B7, which had the highest antifungal rate against *Aspergillus*, was selected as the antagonist by the dual culture method (Figure 2A). The cells of strain B7 stained positively with Gram stain and were rod-shaped (Figure 2B). The results of physiological and biochemical tests are shown in Table 1. The type of oxidative fermentation of glucose was fermentation type; for salt tolerance, it was negative under 5% and 10% conditions, and under 2% conditions, it was positive and suitable for strain growth; sodium citrate utilization was negative, and the rest of the 10 indexes, including the V-P test and the starch hydrolysis test, were positive. Combined with the morphological characteristics of strain B7, guided by these tests, it was initially hypothesized that strain B7 belonged to the genus *Bacillus*. Plotting the growth curve of antagonistic bacteria by OD_600_ absorbance values (Appendix A), the logarithmic growth phase began at 4 h, and the number of *Bacillus* cells grew exponentially until 14 h. The obtained B7 16S rDNA gene sequences were analyzed using NCBI to Blast. The B7 strains showed more than 99% homology with *Paenibacillus polymyxa* IIHR GLFB02 in the NCBI database. The phylogenetic tree was constructed as shown in Figure 3. The antagonist strain B7 was identified as *Paenibacillus polymyxa* based on morphological characteristics and molecular biology. Figure 4 shows that the inhibition rate was 48%. Infection of *A. tubingensis* by *P. polymyxa* occurred with severe rupture of the mycelium and less change in the sporocarp, which became visibly wrinkled (Figure 5).

### 2.3. Analysis of Protein Concentration, SDS-PAGE Electrophoresis Result of Antifungal Proteins, and SEM Evaluation of the Inhibitory Effect of Antifungal Proteins on A. tubingensis Mycelial Growth

The protein standard concentration data were obtained from the enzyme marker, and then the graphic (standard curve) was plotted (Appendix A). With the help of a standard curve, the measured protein concentration of the sample was obtained as 1.43 mg/mL. The crude protein with antifungal activity was obtained by ammonium sulfate precipitation. Several protein fractions had protein molecular masses of about 37 kDa and 60 kDa (Figure 6). The results were similar to the electrophoretic results of Ran [31].

As can be seen in Figure 7, after treatment with antifungal proteins, the mycelial morphology of *A. tubingensis* shriveled and underwent significant breakage and fragmentation. Similarly, an antifungal protein produced by *P. polymyxa* strain VLB16 caused severe alterations in cell morphogenesis that gave rise to swollen and rounded hyphae [37].

### 2.4. Effect of P. polymyxa on the Transcriptome of A. tubingensis

#### 2.4.1. Analysis of Differentially Expressed Genes (DEGs)

In order to investigate the molecular mechanism of the action of *P. polymyxa* B7 on *A. tubingensis* A1, RNA sequencing was carried out in a co-culture of *P. polymyxa* B7 with *A. tubingensis* A1 in a sterile environment. The results showed that several genes of *A. tubingensis* were differentially expressed under the biological regulation of *P. polymyxa*. The DEGs in the control and treatment groups are shown in Figure 8. Each point in the figure represents a gene, and yellow dots, red dots, and blue dots represent significantly downregulated, upregulated, and non-significantly DEGs, respectively. A total of 1183 DEGs were screened using |log2 (Fold Change)| ≥ 2 and FDR < 0.05 as criteria, 883 were upregulated expressions, and 300 were downregulated expressions. The downregulated expressions of the differential genes indicated the inhibitory effect of *P. polymyxa* treatment on the growth of *A. tubingensis* A1, which provided a prerequisite for further research on the inhibitory mechanism of *P. polymyxa* on *A. tubingensis* A1.

#### 2.4.2. GO Enrichment Analysis of DEGs

All genes and DEGs were categorized by the degree of GO enrichment into three major categories: biological process, molecular function, and cellular component, which contained 27, 17, and 3 secondary functions, respectively, with a certain number of DEGs under each function. The highest number of differential genes in biological processes was found in the cellular process (625), metabolic process (593), and biological regulation (276), respectively, mainly enriched in functions related to the organonitrogen compound biosynthetic process (GO: 1901566). The highest number of differential genes in molecular functions was found in binding (499) and catalytic activity (479), which were mainly enriched in functions related to nucleic acid binding (GO: 0003676). The highest number of differential genes in cellular components was found in the cellular anatomical entity (503) and protein-containing complex (308) mainly enriched in functions related to non-membrane-bounded organelles (GO: 0043228), respectively (Figure 9).

#### 2.4.3. KEGG Enrichment Analysis of DEGs

The KEGG enrichment classification of differential genes showed a total of 605 differential genes involved in 101 metabolic pathways. The enrichment results in Appendix A showed that the pathways involving many upregulated genes mainly included metabolic pathways and ribosomes. Figure 10 shows that differential genes were mainly involved in five categories of primary metabolic pathways, namely, metabolism, genetic information processing, cellular processes, environmental information processing, and organismal systems, and these primary metabolic pathways were divided into a total of 19 secondary metabolic pathways. KEGG pathway annotations indicated that differential genes were mainly involved in different metabolic processes such as metabolic processes of amino acids, sugars, lipids, energy, and so on. Among the top 20 pathways with the most significant enrichment, metabolic pathways and biosynthesis of secondary metabolites in eukaryotes were the most significant. The metabolic pathway genes OtaA, LDHA, Erg-6, GST, al-1, dapA, PDH1, alcA, DAS1, and csn were significantly downregulated (Table 2). To investigate the molecular mechanism of secondary metabolite synthesis, its biosynthesis-related genes were screened based on transcriptome sequencing results. Table 3 shows that iolG4, CIMG_05755, mug58, notG, bglF, qutE, and icl1 were significantly downregulated.

#### 2.4.4. Real-Time Quantitative PCR (RT-qPCR) Validation of DEGs

The expression levels of these genes were detected by RT-qPCR, as shown in Figure 11. The expression levels of the genes RPL6, RPS8A, and RPS9 were upregulated, while the expression levels of the genes blh, CDO1, PepA, RntA, and YAT1 were downregulated, which was basically consistent with the results of transcriptional analysis and proved that the transcriptome data were reliable.

## 3. Discussion

It was found that *Alternaria*, *Aspergillus*, *Cladosporium*, *Papiliotrema*, *Wallemi*, and *Ustilaginoidea* were the most dominant fungal genera in paddy grain at different storage periods [44]. Among them, *Aspergillus* was the dominant fungal genera in the paddy microbial community at different storage temperatures and relative humidities [42]. The frequency of isolation of *Aspergillus* in paddy in Qatar was 78.57% [8]. The relative abundance of *Aspergillus* increased in the later stages of the storage phase and decreased with increasing storage depth [43]. There were significant differences in the fungal communities of *Aspergillus* in paddy grains at different storage stages. The dominant fungi in paddy grains from southern China were mainly *Aspergillus flavus* and *Aspergillus niger* [43]. *Aspergillus niger* aggregates isolated from Spanish paddy grains were most abundant in *Aspergillus tubingensis* [45].

Recently, biocontrol using microorganisms isolated from the environment has become an important means of reducing the use of chemicals. *Bacillus amyloliquefaciens* BUZ-14 has been found to be a very suitable biocontrol agent for application during storage [46]. *Bacillus subtilis* F3 is an antifungal bacteria that can antagonize a wide range of pathogens [47]. In our study, *P. polymyxa* B7 was selected for its inhibitory effect on *Aspergillus* spp. fungi of storage paddy. *P.polymyxa* NX20 inhibited the growth of *Phytophthora capsici* mycelium [48], and *P. polymyxa* YF had an inhibition rate of 70.69% against the mycelial growth of *F. avenaceum*. Abnormal branching, blister deformity, and cell wall rupture of *F. avenaceum* mycelium occurred after treatment with fermentation broth of *P. polymyxa* [49]. *P. polymyxa* AF01 was able to directly inhibit mycelial growth, spore germination, and shoot tube elongation, causing irreversible damage to the integrity of the cell membrane and cellular ultrastructure [50].

Antifungal proteins purified from *P. polymyxa* VLB16 inhibited the growth of *Pyricularia grisea* and *Rhizoctonia solani*, and SDS-PAGE electrophoresis showed that the molecular weight of the protein was 37 kDa [37]. A protein of about 71.9 kDa with broad-spectrum antifungal activity was isolated from JSa-9 cultures [51]. A 36 kDa protein with broad-spectrum antifungal activity was purified from the *P. polymyxa* 7F1 [31]. The antifungal protein caused severe alterations in cellular morphogenesis resulting in swollen and rounded hyphae [37]. In our study, the antifungal proteins produced by *P. polymyxa* B7 have a disruptive effect on the growth of *A. tubingensis* mycelia.

Through SEM observation, it was found that the mycelium of *A. tubingensis* treated with *P. polymyxa* underwent folding and rupture, and the results showed that *P. polymyxa* had a destructive effect on the growth of *A. tubingensis* mycelium, suggesting that its antifungal activity was related to the downregulation of some key synthetic genes. To address the above issues, we analyzed the significantly downregulated differential genes and analyzed their metabolic pathways. Riboflavin metabolism, Tryptophan metabolism, and Cysteine and methionine metabolism-related functions were downregulated based on differential gene analysis. RNA-seq analysis showed 280 upregulated genes and 333 downregulated genes. Functional and GO enrichment analyses showed the downregulation of processes related to metabolic and oxidative processes as well as functions related to oxidoreductase and hydrolase activities [52].

OtaA is the initiating enzyme for OTA synthesis, which utilizes acetyl coenzyme A and malonyl coenzyme A to synthesize 7-methylhoney monotetrasporine, which is then oxidized into OTβ catalyzed by OtaC [53]. Downregulation of the OtaA gene decreases the synthesis of OTA to some extent. Studies have shown that LDHA promotes glycolysis and TCA cycling and that LDHA regulates cell differentiation by modulating cellular NAD^+^ levels and subsequent downstream effects on mitochondrial function [54]. Downregulation of LDHA may inhibit cell differentiation. Lipid droplets (LDs) are cytoplasmic lipid storage organelles that play a role in lipid metabolism, transport, and signaling. Erg6 is an LD marker [55]. Downregulation of Erg6 affects lipid metabolism, transport, and signaling in fungi. The Erg6 gene encodes 24-SMT, ∆24-sterol methyltransferase (24-SMT), a key enzyme that plays an important role in the ergosterol biosynthesis pathway of fungi and inhibits the growth of *Aspergillus fumigatus* Af239 by affecting the 24-SMT [56]. Downregulation of the erg6 gene inhibited the growth of *A. tubingensis* to some extent. The results of transcriptome analysis showed that *P. polymyxa* treatment may affect plasma membrane integrity, transmembrane transport, energy metabolism, and signaling in *Phytophthora capsici* [48]. Glutathione-S-transferase (GST) is mainly involved in the regulation of intracellular redox reactions and detoxification metabolism, and GST transporter proteins are involved in the metabolism of 6:2 FTCA, and the reduction in metabolizing enzyme activities reflects the impaired detoxification system of the cell [57].

## 4. Materials and Methods

### 4.1. Sample Source and Species

Our sample source was the grain reserve warehouse of Jiangsu Runguo Agricultural Development Co., Ltd., Zhenjiang city, Jiangsu Province, China, paddy field. Japonica rice variety Nan Jing 46 was the tested paddy variety.

### 4.2. Isolation and Purification of Dominant Molds

Paddy grains of different storage periods obtained in batches from the grain reserve warehouse of Jiangsu Runguo Agricultural Development Co., Ltd. were put into sealed bags and transported back to the laboratory and then suspected moldy paddy grains were selected to isolate the fungi. After weighing 5 g of paddy, it was disinfected with 15 mL of NaClO solution for 3 min and then washed with 10 mL of sterile distilled water (SDW) three times. Then, 10 mL of SDW was added to prepare the gradient solution, and 100 µL of gradient solution was taken and applied to Bengal red and culture medium and incubated for 7 d at 28 °C. The different types of strains were counted, and the potato dextrose agar (PDA) delineation separation method was used on it several times to purify the strains. To confirm the dominant strains, the purified strains were inoculated in PDB medium containing 15% glycerol and stored at −80 °C for a long time.

The dominant molds were isolated from the surface of moldy paddy at different storage periods and identified by morphological characteristics and DNA sequencing. They were incubated on the surface of PDA plates at 28 °C for 5 d and stored at 4 °C.

### 4.3. Isolation and Characterization of Antagonist Strains

#### 4.3.1. Isolation of Antagonist Bacteria (*P. polymyxa* B7)

The source was healthy paddy inter-root soil from paddy fields in Zhenjiang city. The weight of the soil sample consisted of 10 g of paddy rhizosphere soil, and 15 mL of SDW was added into rotary shaker (180 rpm) for 20 min to obtain the stock solution. Then, 100 µL of dilution was applied to PDA medium and cultured at 28 °C for 7 d. The strains with inhibitory effect were screened out, and the isolated and purified strains were inoculated into the LB medium, at 28 °C, with the rotary shaker (180 rpm), and inoculated for 24 h. The samples were collected and stored at 4 °C.

The dual culture method evaluated the inhibitory effect of the isolated strains. Holes were punched in the center of PDA plates (90 mm) with a diameter of 6 mm, about 20 mm from the center hole, spaced 4 holes apart, and the agar in the holes was removed. Different dominant mold spore suspensions of 30 µL 1 × 10^5^ cfu/mL and antagonist suspensions of 30 µL 1 × 10^7^ cfu/mL were selected for the test and incubated at 28 °C for 5 days. Then, the size of the pathogenic fungal hyphae diameter was observed and measured. Three replicates were analyzed for each treatment.

#### 4.3.2. Determination of Growth Curves of *P. polymyxa* B7

Fifty-two test tubes, each containing 10 mL of LB liquid medium, were taken and incubated for 24 h with 1% inoculum. Tubes were cultivated at 28 °C and 180 rpm under the rotary shaker. Samples were taken every two hours during the 26 h incubation period and placed in the refrigerator. After all the incubations, the samples were diluted, and the absorbance values at OD_600_ were determined uniformly.

#### 4.3.3. Physiological, Biochemical, and Molecular Biological Characterization of *P. polymyxa* B7

The bacteria with the best antagonistic effect were analyzed. Morphological characteristics of strain B7 were observed and recorded after 24 h of incubation at 28 °C on LA agar. Physiological and biochemical tests for bacterial identification of strain B7 according to the *Handbook of Identification of Common Bacteria* [58]. The 16S rDNA gene was amplified using universal primers 27F (5′-AGAGTTTGATCCTGGCTCAG-3′) and 1492R (5′-GGTTACCTTGTTACGACTT-3′) [59]. PCR products were sequenced at Sangon Biotech Co., Ltd. (Shanghai, China), and NCBI (https://www.ncbi.nlm.nih.gov/ (accessed on 15 August 2022)) was used to blast the sequencing results. The related sequences were downloaded, and the phylogenetic tree was constructed using MEGA 11.

### 4.4. Inhibitory Effect of P. polymyxa B7 on A. tubingensis A1

The inhibitory effect of the isolates on *A. tubingensis* A1 was determined by the dual culture method. According to the method in 4.3.1, four 20 mm equally spaced holes were made in the center of the plate, and the agar in the holes was removed. Then, 1 × 10^5^ cfu/mL of fungal spore suspension was put into the center of the hole, and the two holes on the left and right were inoculated with 1 × 10^7^ cfu/ml of the suspension of the strain, and the other two holes were left blank as a control. The plates were placed in the 28 °C incubator for 5 d and contact between the strain and the pathogenic bacterial hyphae mycelium was observed. The size of the mycelium diameter of the pathogen was measured according to the crossover method and calculated according to the following formula:Inhibition rate (%) = (Control mycelium diameter-Mycelium diameter of treatment group/Control mycelium diameter) × 100%(1)

The rate of inhibition was calculated, and each treatment was repeated three times.

### 4.5. Inhibition of A. tubingensis A1 Mycelial Growth Caused by P. polymyxa B7

According to the method of Qian Zhang [60] and the method in 4.3.1, the two holes on the left and right were inoculated with 100 µL 1 × 10^7^ cfu/mL of the suspension of the strain, and then Φ6 mm fungus cube of *A. tubingensis* was placed in the center of the medium, and an equal amount of SDW was applied as a control. Three replicates were performed for each treatment. They were incubated at a temperature of 28 °C for 3 d. According to the results of the dual culture method in 4.4, a Φ8 mm coverslip was inserted obliquely around the inhibitory circle to promote mycelial growth, which was recorded as the T-treatment group, and the normal-growing mycelium of *A. tubingensis* was recorded as the CK group. According to the method of Ye [61], with slight modification, CK and T-treatment groups were fixed with 2.5% glutaraldehyde solution (Sinopharm, China) at 4 °C and without light for 24 h. Each coverslip placed in the 24-well plates was dehydrated using ethanol at different concentrations (30%, 50%, 70%, 85%, 90%, 95%, and 100%) for 15 min, and finally 1 mL of tert-butanol (replacing the ethanol) was added to each well. Three repetitions of the treatment and then freeze-drying were performed, and the drying was ended when the pressure in the freeze-dryer decreased to 10 Pa. The samples were treated with gold spraying and then observed using field emission scanning electron microscope (FESEM) (JSM-7800F).

### 4.6. Extraction of Antifungal Substances of P. polymyxa B7

#### 4.6.1. Seed Fermentation Broth Preparation

Single colonies of the strain cultured for 24 h were picked out and added to 50 mL (100 mL conical flask) of LB liquid medium. The culture was incubated at 28 °C and 180 rpm in a rotary shaker. The length of incubation time was determined according to the growth curve, and the seed fermentation broth was obtained.

#### 4.6.2. Preparation of Bacteria-Free Culture Solution of the Strain

The seed fermentation broth of the strain in logarithmic growth phase was inoculated into 100 mL (250 mL conical flask) of LB liquid medium at 1% inoculum volume and incubated for 72 h at 28 °C in a rotary shaker at 180 rpm to obtain the fermentation broth. The fermentation broth was transferred to a centrifuge tube and centrifuged at 4 °C and 10,000 rpm for 15 min to remove the bacteria. The supernatant was filtered through a 0.22 µm diameter aqueous bacterial membrane to produce a bacteria-free culture solution and stored in a 50 mL centrifuge tube in the refrigerator.

#### 4.6.3. Crude Extraction of Antifungal Proteins

Solid ammonium sulfate was added to the supernatant of bacteria-free culture solution to 20%, 30%, 40%, 60%, and 80% saturation, and the inhibitory crude proteins precipitated by ammonium sulfate at different saturation levels were precipitated. The antibacterial activity was determined by the dual culture method with 50 µL of bacteria-free culture solution per well, and the antifungal activity of the precipitated crude proteins with different saturation levels was compared to determine the optimal concentration of ammonium sulfate.

Based on the above results, solid ammonium sulfate was slowly added to the bacteria-free fermentation supernatant to saturate it to an optimal inhibitory concentration of 40%, and it was set aside at 4 °C for 24 h. The precipitate was collected by centrifugation at 1650 g (rcf) for 15 min. The resulting precipitate was fully solubilized in 2 mmol/L phosphate buffer (pH 7.2). The dissolved solution was transferred to a dialysis bag with a molecular weight cut-off of 8–10 k Da for 48 h. During this period, the dialysis solution was replaced 4 times, and the liquid in the bag was filtered through a 0.22 µm filter, and the filtrate was the crude extract of antifungal proteins. The crude extract of antifungal proteins was stored at −20 °C for overnight freezing, and then freeze-dried in a vacuum freeze-dryer for 48 h. Antifungal proteins in white powder form were obtained.

### 4.7. Determination of Inhibitory Protein Concentration and SDS-PAGE Electrophoresis

According to the Bradford Protein Assay Kit, protein standards were configured by dissolving the samples through PBS buffer. Firstly, 5 μL of protein standards of different concentrations were taken and added to the protein standard wells of the 96-well plate. Then, 5 μL of samples was taken into other sample wells of the 96-well plate. If the sample was less than 5 μL, standard diluent was added until the volume reached 5 μL. Secondly, 250 μL of G250 staining solution was added to each well. At last, A595 was determined using an enzyme marker, and then the standard curve and the sample volume were used to calculate the protein concentration.

Antifungal protein sodium dodecyl sulfate—polyacrylamide gel electrophoresis (SDS-PAGE): the protein samples were mixed with the sampling buffer (5×) at 4:1, and the water bath was boiled for 5 min and then cooled to room temperature. In total, 10% separating gel and 5% concentration gel were used, and the protein gel was stained with Coomassie Brilliant Blue staining solution for 30 min after electrophoresis. Then, a decolorization shaking machine was used with decolorization solution to decolorize the protein gel, until protein bands were clearly visible.

### 4.8. Effect of Antifungal Proteins on the Growth of A. tubingensis A1 Mycelia and SEM Observations

A suspension of *A. tubingensis* spores at 1 × 10^6^ cfu/mL was added to PDB culture medium with an antifungal protein concentration of 100 µg/mL. PDB medium without antifungal protein was used as a control. *A. tubingensis* was incubated at 28 °C for 38 h and filtered through filter paper to obtain mycelia. According to the method of Li [19], the samples were processed. After ethanol dehydration of the mycelium was finished step by step, 1 mL of tert-butanol solution was added to treat the sample 3 times, each time for 15 min. A small amount of tert-butanol was left to vacuum freeze-dry, and the drying was ended when the isobaric pressure was 10 Pa. The mycelia were glued to the sample stage. After gold spraying, the sample was observed and photographed under FESEM (JSM-7800F).

### 4.9. Transcriptomics of P. polymyxa B7 against A. tubingensis A1

#### 4.9.1. Sample Preparation and RNA Extraction

In total, 100 µL of *A. tubingensis* spore suspension (diluted to a concentration of 1 × 10^5^ cfu/mL) was inoculated into 50 mL of PDB medium, and after 18 h of incubation in a rotary shaker at 28 °C, 180 rpm, and without light, the inoculum was inoculated into new PDB medium according to the same inoculum amount and incubated for 18 h under the same conditions. *P. polymyxa* B7 was activated in LB medium and incubated at 28 °C and 180 rpm in a rotary shaker without light for 18 h. After that, it was inoculated into the new LB medium according to the 1% inoculum and incubated for 18 h under the same conditions.

The two kinds of microorganisms above were centrifuged, and the collected body was dissolved in 100 mL of PDB medium, and the control group was the same amount of *A. tubingensis* dissolved in 100 mL of PDB medium. Three replicates were made for all the above samples (CK1, CK2, CK3, T1, T2, and T3, respectively). After incubation in a rotary shaker at 28 °C and 180 rpm for 10 h without light, the organisms were collected by centrifugation at 410 g (rcf) for 5 min, washed three times with PBS buffer, and then the samples were pre-cooled with liquid nitrogen and stored in an ultra-low temperature refrigerator at −80 °C. The total RNA was extracted according to the instructions of Trizol reagent kit (Invitrogen, Carlsbad, CA, USA), and the concentration, purity, and integrity of the RNA in the extracted samples were detected using an Agilent 2100 Bioanalyzer (Agilent Technologies, Palo Alto, CA, USA) and checked using RNase free agarose gel electrophoresis.

#### 4.9.2. Transcriptome Sequencing and Bioinformatics Analysis

The qualified RNA samples were sent to Guangzhou Gene Denovo Co., Ltd. (Guangzhou, China) for cDNA library construction, library quality testing, and high-throughput sequencing on Illumina HiSeq 2500. mRNA was enriched by removing the rRNA, reverse transcribing the enriched mRNA to form a double-stranded cDNA, repairing the cDNA double-end and splicing, and then PCR amplification was performed to realize cDNA library construction. After sequencing, the sequencing data were filtered, and the raw reads were filtered by fastp for quality control, and the clean reads were obtained by filtering the low-quality data, which were then aligned with the reference genome of *A. tubingensis* for a series of bioinformatics analyses. (ASM1334032v1 of *A. tubingensis* was selected as the reference genome.)

DESeq2, a method for differential analysis of count data, was used for shrinkage estimation for dispersions and fold changes to improve stability and interpretability of estimates. RNA differential expression analysis was performed by DESeq2 3.11 [62] software between two different groups (and by edgeR [63] between two samples). Genetic parameters with false discovery rate (FDR) ≤ 0.05 and absolute fold change (FC) ≥ 4 were considered as DEGs [64]. The expressed genes (FDR ≤ 0.05, absolute FC ≥ 4) were functionally annotated according to Kyoto Encyclopedia of Genes and Genomes (KEGG) and Gene Ontology (GO) databases [65]. In addition, both upregulation and downregulation of gene expression profiles were analyzed for GO and KEGG enrichment.

#### 4.9.3. RT-qPCR Validation

Genes with significant differences were selected from the obtained transcriptome structures. Ubiquitin gene was used as a reference gene for relevant gene expression analysis. The primers used are shown in Table 4. A 20 µL reaction system was used, and the following parameters were set in a RT-qPCR instrument: pre-denaturation at 95 °C for 30 s and denaturation at 95 °C for 10 s, followed by annealing at 60 °C for 30 s, and finally extension at 72 °C for 20 s, for 40 cycles; the dissolution curves were 95 °C for 15 s, 60 °C for 1 min, and 95 °C for 15 s.

### 4.10. Statistical Analysis

Average comparison of means and treatment group scores were obtained using Duncan’s Multiple Range Test. Differences were considered to be significant at *p* ≤ 0.05. All experimental data were statistically analyzed using Excel 2010, SPSS 22.0, and Origin 2019b. Transcriptome bioinformatics were analyzed using OmicShare (https://www.omicshare.com/, accessed on 25 October 2023). At least three replicate measurements were used for each treatment.

## 5. Conclusions

In this study, the screening of *Bacillus* for its antifungal effects on *Aspergillus* was carried out to obtain *P. polymyxa*. Observed by SEM, the mycelial growth of *A. tubingensis* was ruptured and fragmented after treatment by *P. polymyxa* B7. *A. tubingensis* mycelial growth was inhibited by the treatment of the antifungal proteins (secondary metabolites) produced from *Bacillus*, and the antifungal proteins proved to have the potential to control *A. tubingensis* contamination. The expression of *A. tubingensis* metabolism-related genes was downregulated under *P. polymyxa* treatment, affecting its normal physiological metabolism and the synthesis of secondary metabolites, as shown in transcriptome analysis. The results of the study provide a new strategy for the prevention of *A. tubingensis* contamination to paddy, therefore safeguarding food security and decreasing paddy losses.

## Figures and Tables

**Figure 1 ijms-25-02195-f001:**
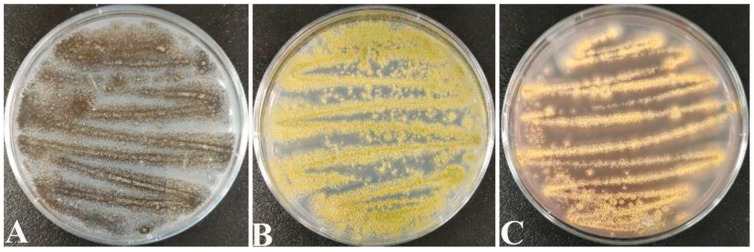
(**A**) *Aspergillus tubingensis*; (**B**) *Aspergillus flavus*; (**C**) *Aspergillus ochraceus*.

**Figure 2 ijms-25-02195-f002:**
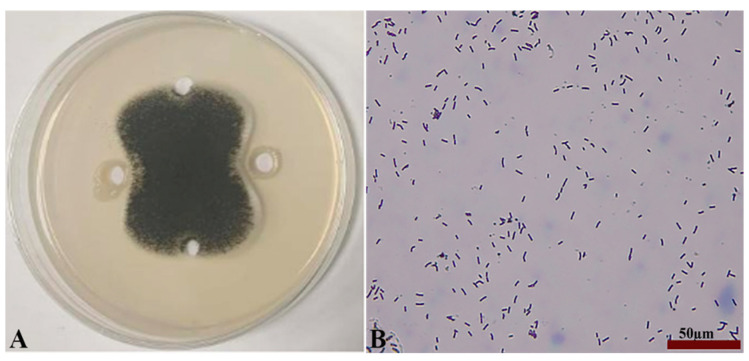
Inhibition of *Aspergillus* by strain B7 (**A**); Gram staining results and its morphology (**B**), the scale in the figure is 50 µm.

**Figure 3 ijms-25-02195-f003:**
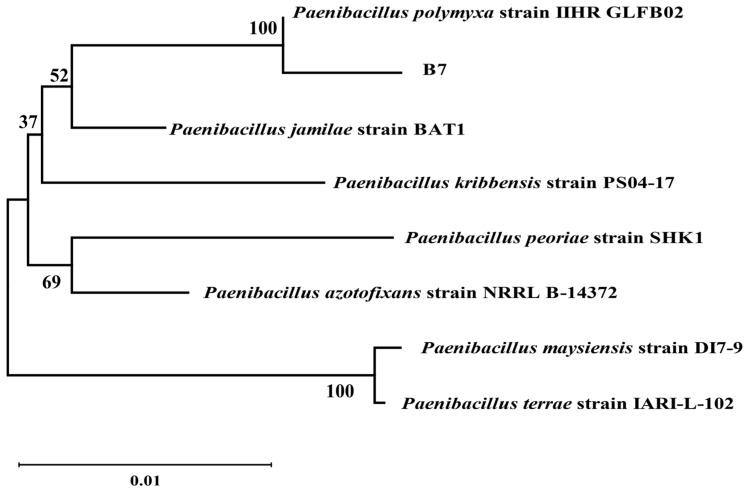
Phylogenetic tree of strain B7. The obtained B7 16S rDNA gene sequences were analyzed using NCBI to Blast. The phylogenetic tree was constructed by neighbor-joining method using MEGA11. Bootstrap values are shown at branch points.

**Figure 4 ijms-25-02195-f004:**
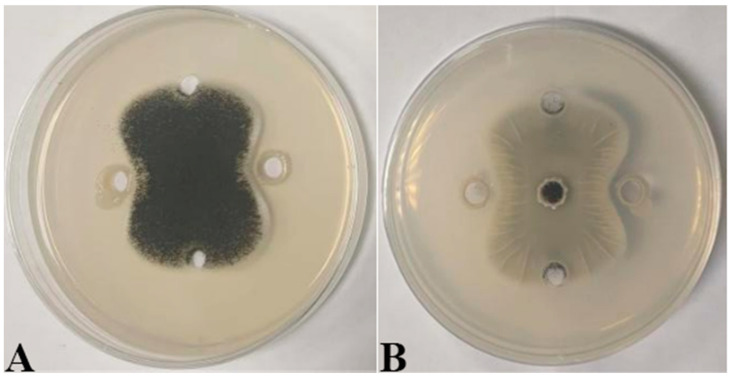
Inhibitory effect of *P. polymyxa* B7 on mycelial growth of *A. tubingensis* A1: (**A**) front; (**B**) the other side.

**Figure 5 ijms-25-02195-f005:**
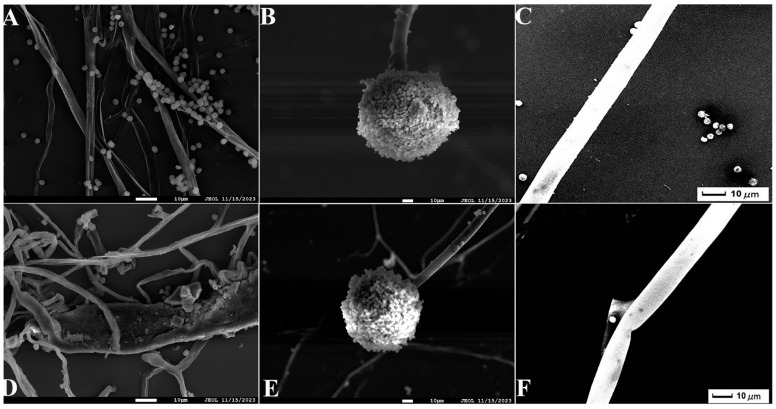
Scanning electron microscope (SEM) observation of *A. tubingensis* mycelia treated with *P. polymyxa*: (**A**) the control mycelia (1000×); (**B**) the control sporangia (500×); (**C**) the control sporophore (1000×); (**D**) the treatment mycelia (1000×); (**E**) the treatment sporophore (500×); (**F**) the treatment sporophore (1000×). The scales in the figures are 10 µm.

**Figure 6 ijms-25-02195-f006:**
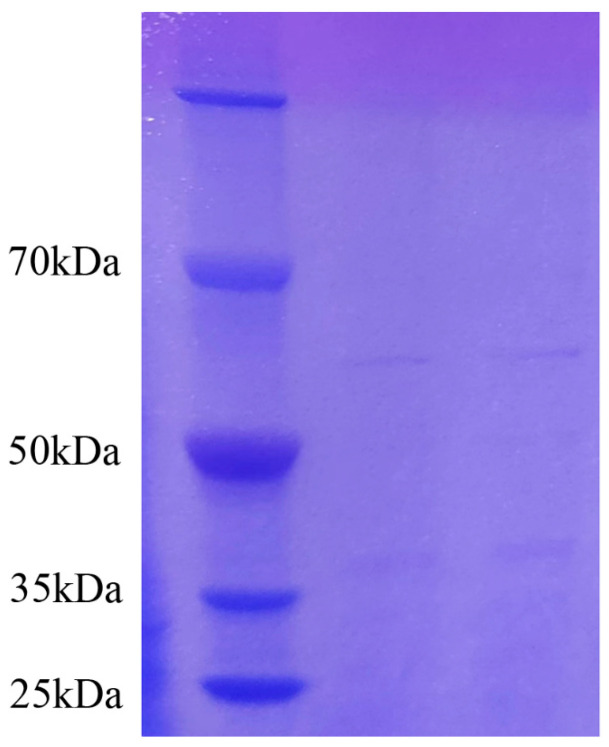
The SDS-PAGE electrophoresis result of antifungal crude protein (the left lane is the marker, and the other lanes show the antifungal crude protein). The crude protein with antifungal activity was obtained by ammonium sulfate precipitation.

**Figure 7 ijms-25-02195-f007:**
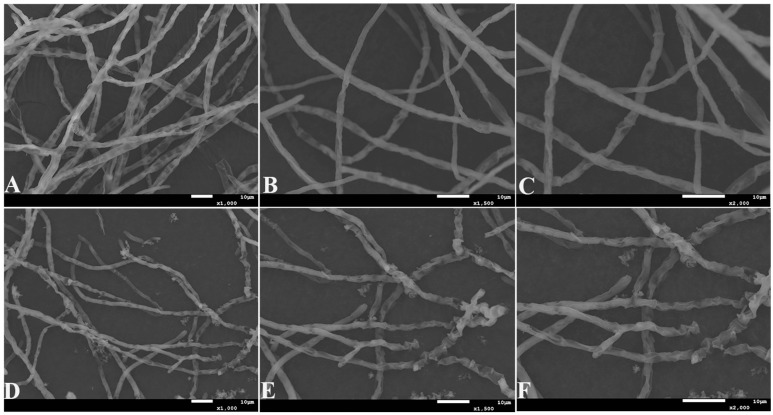
(**A**) *A. tubingensis* mycelia treated without the addition of inhibitory proteins (1000×); (**B**) *A. tubingensis* mycelia treated without the addition of inhibitory proteins (1500×); (**C**) *A. tubingensis* mycelia treated without the addition of inhibitory proteins (2000×); (**D**) *A. tubingensis* mycelia treated with the addition of inhibitory proteins (1000×); (**E**) *A. tubingensis* mycelia treated with the addition of inhibitory proteins (1500×); (**F**) *A. tubingensis* mycelia treated with the addition of inhibitory proteins (2000×).

**Figure 8 ijms-25-02195-f008:**
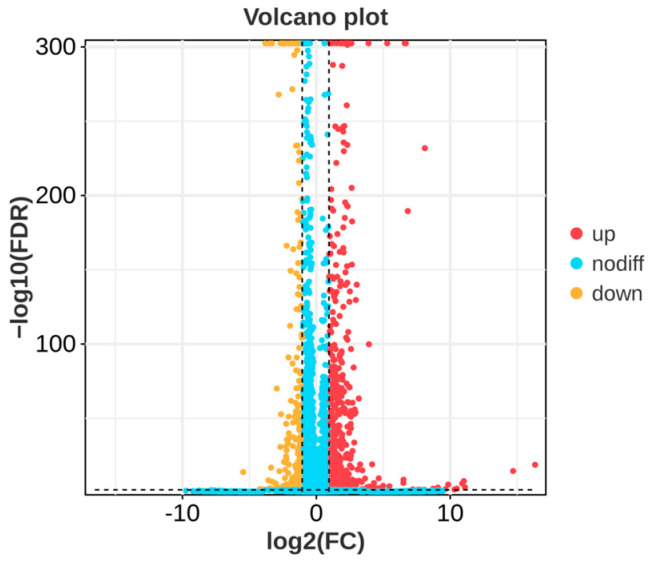
Volcano plot of differentially expressed genes (DEGs).

**Figure 9 ijms-25-02195-f009:**
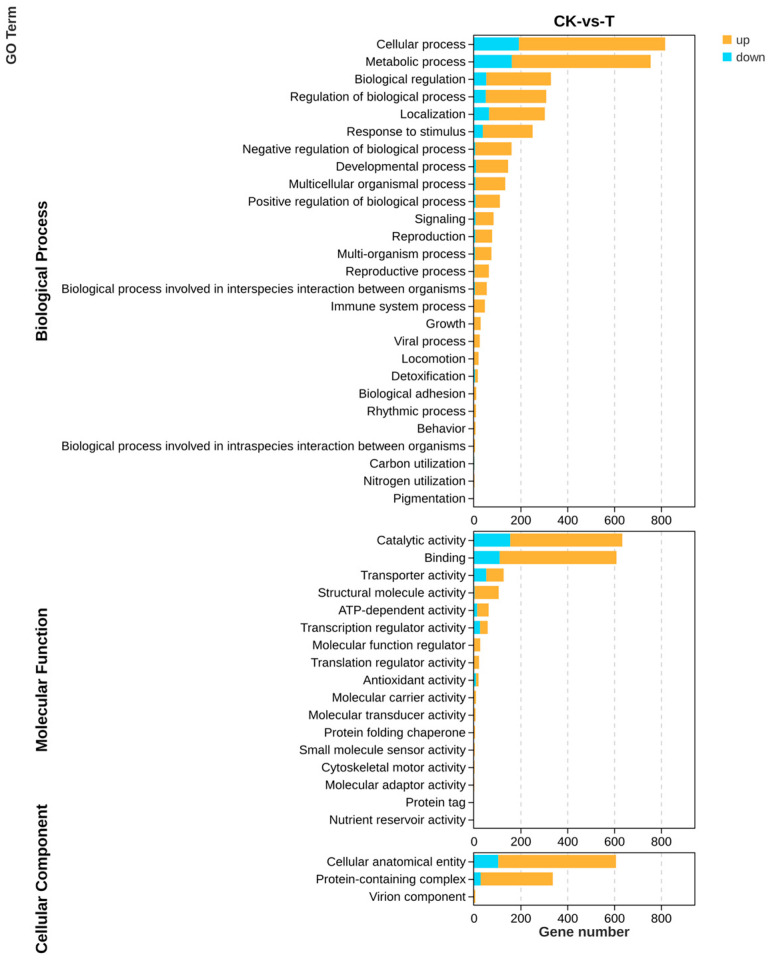
GO analysis of differential genes of the transcriptome about the effect of *P. polymyxa* on *A. tubingensis*.

**Figure 10 ijms-25-02195-f010:**
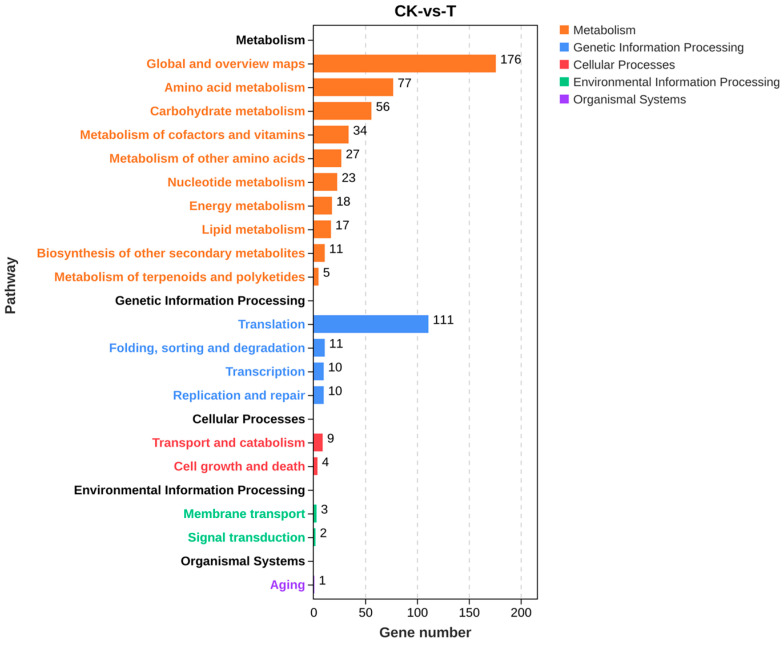
KEGG pathway classification of the screened differential genes.

**Figure 11 ijms-25-02195-f011:**
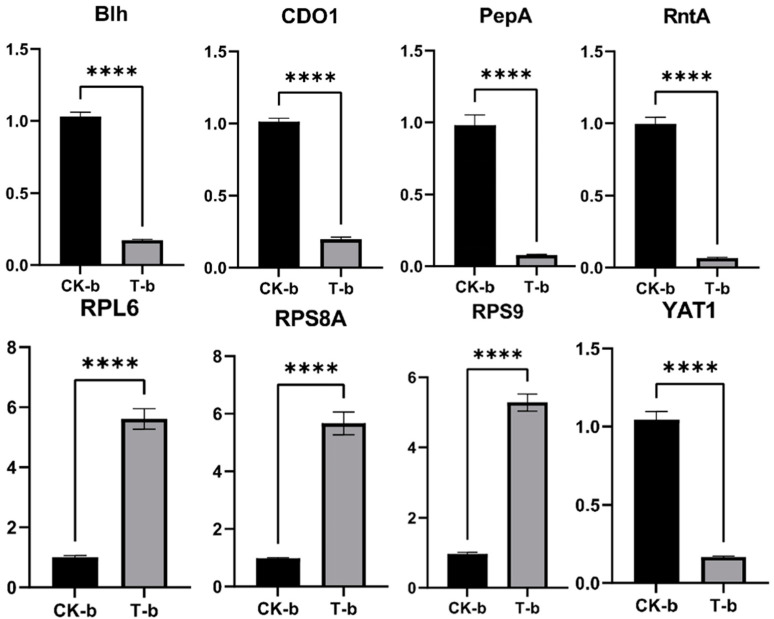
Validation of some DEGs by real-time quantitative PCR (RT-qPCR). ****: *p* < 0.0001.

**Table 1 ijms-25-02195-t001:** Physiological and biochemical characteristics of strain B7.

Test Item	Result	Test Item	Result
Salt resistance		Glucose oxidative fermentation	Fermentation type
2%	+	Sodium citrate utilization	−
5%	−	Sugar fermentation	
10%	−	D-glucose	+
Exposure to enzymes	+	Mannitol	+
Methyl red	+	Lactose	+
Starch hydrolysis	+	Sucrose	+
V-P assay	+	Gram	+

Note: “+” indicates positive; “−” indicates negative.

**Table 2 ijms-25-02195-t002:** Expression profiling of genes related to metabolic pathways.

Gene ID	Gene Description	Log_2_FC
ncbi_56006507	Ornithine aminotransferase	−7.564785
ncbi_56002670	L-lactate/malate dehydrogenase	−7.494522
ncbi_56004126	Squalene synthetase	−7.400879
ncbi_56006177	Glutathione S-transferase	−6.853829
ncbi_56007485	Phytoene dehydrogenase	−5.403722
ncbi_56006167	Dihydrodipicolinate synthetase family protein	−5.357552
ncbi_56001048	Phytoene dehydrogenase	−4.459432
ncbi_56003329	Threonine aldolase	−4.058894
ncbi_56005832	Dihydroxy-acetone synthase	−4.058894
ncbi_56002418	Fungal chitosanase	−3.984418

**Table 3 ijms-25-02195-t003:** Expression profiling of genes related to secondary metabolite synthesis.

Gene ID	Gene Description	Log2FC
ncbi_56007446	NAD binding Rossmann fold oxidoreductase	−3.529698
ncbi_56002881	Fructose-bisphosphate aldolase	−3.06848
ncbi_56001061	D-glycerate 3-kinase-like protein	−2.992715
ncbi_56007372	Cytochrome P450	−2.907936
ncbi_56006356	Beta glucosidase	−2.898623
ncbi_56008280	Catabolic 3-dehydroquinase	−2.896164
ncbi_56005213	Isocitrate lyase	−2.069498

**Table 4 ijms-25-02195-t004:** Primer design table for differential genes.

Gene ID	Symbol	Primers (5′ to 3′)
ncbi_56000074	UFD1	F: CGGGTGATGTCCAGTGTAAG
		R: GGTGCCATAAGCCGAGTC
ncbi_56002166	UEF-CFP	F: GCGAAGAAAAGGGTCAAAG
		R: GGTAATGTCAAGGCTGGTCA
ncbi_56000480	PepA	F: GTTGAGCGGCTACTCTTGG
		R: AGGTGGTCTGTGCCTTGG
ncbi_56000968	EcdD	F: GGCTATCATCTTGGGAGGTG
		R: AGTCCGAGTCTTGGGGCT
ncbi_56001045	RntA	F: GAGCATCATCGCCCTTCT
		R: TGCGGGTAGTCGTCAATAGTA
ncbi_56002623	Blh	F: TCAACAGAATCAAGCCCCA
		R: CGAGTCGCCCGTGAATAC
ncbi_56002624	CDO1	F: CAACTTTCCCTGGCTCTGC
		R: AACCTTGTGGTCAACATTTTCT
ncbi_56008178	YAT1	F: CATTGAATCGCCCAAGTCT
		R: CGAAGCCAAAGTGCCGTA
ncbi_56004358	RPS9	F: CGCAACAAGCGTGAGGTG
		R: CAGACGGCGAATCAAAGC
ncbi_56004736	RPS8A	F: AACTTCTCGTGGGGTTCTGA
		R: GGTCTCGGTCTTCTGCTGG
ncbi_56008317	RPL6	F: ATGTCGGACTCTACTGTTGGC
		R: CGGACTTTCTTGGGCTGC

## Data Availability

The data shown in this study are included in the article.

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
