# Peer review of "Isolation and Characterization of *Paenibacillus polymyxa* B7 and Inhibition of *Aspergillus tubingensis* A1 by Its Antifungal Substances"

_ijms, 2024, doi:10.3390/ijms25042195_

Round 1

Reviewer 1 Report

Comments and Suggestions for Authors

The authors' article is devoted to a current problem in the field of agriculture: screening of Paenibacillus polymyxa B7 and inhibition of As-pergillus Tubeingensis with its antifungal substances. This issue is clearly of interest to scientists working in this area. The experimental design is suitable, however, it is necessary to change the “results” section, since it is not always clear from the text of the article which technique was used to obtain a particular result. In addition, it is necessary to add “connecting” data when moving from one section to another so that the logic of this work can be more clearly traced. In addition, I have several recommendations for authors:

1. Statements in lines 31-35, 41-45 require confirmation (literature references)

2.  It is necessary to expand the “Introduction” section with data from previously conducted similar studies. It is necessary to add information about the importance of the study, not only from the point of view of economic benefits.

3. The authors need to clarify which statistical processing method was used for which stage of the study

4. Figures 9 and 10 need to be redone or replaced, since in this form they are difficult to read

5. It is necessary to expand the "Conclusion" section   with conclusions about the data obtained, prospects for application, etc.

6. The article needs to be checked for typos, for example: it is necessary to remove the hyphen from the title, line 49 "…Other…" is a capital letter necessary? line 63 requires spaces and so on.

7. Please indicate for which type of rice the study was conducted

8. Is Figure 3 necessary in the article?

9.  Sections 2.1 and 2.4 contain too little data and may not need to be highlighted separately.

Comments on the Quality of English Language

Minor editing of English language required

Author Response

Dear Reviewer,

Thank you very much for useful suggestions on our manuscript. It is very kind of you to give us the chance to improve our manuscript. We revised the manuscript carefully according to the suggestions (point by point). And we highlight some changes by using the red text. Please see the attachment.

We would be grate for any suggestions given by you, and we will consider them about. Thank you!

Kind regards,

Yours Sincerely,

Sun Weihong,Zhao Tianyuan 

College of Agricultural Engineering, Jiangsu University

Zhenjiang, Jiangsu Province,212013

P.R. China

Reviewer 2 Report

Comments and Suggestions for Authors

General comments:

The authors present a thorough study of Paenibacillus B7 antagonistic potential against Aspergillus tubingensis a fungal pathogen of rice. The title of the manuscript does not fully represent the content since the authors have concentrated their work on the antimicrobial proteins produced by Paenibacillus and their influence on Aspergillus metabolism. Please update the title to better match the content of the article. I would recommend that the abstract and whole manuscript undergo detailed linguistic corrections. There are sometimes very long sentences that should be separated. Please avoid colloquial phrases. The introduction is very short and does not present the necessary background for the presented study. The results and methods should be described more clearly as it is sometimes hard to understand how the research was carried out. I have not noticed any serious methodological mistakes, only the growth curve analysis methodology seems to include minor methodological problems. For younger cultures it can lead to the overestimation of growth due to growth in lower temperature). For old cultures, the cell density can be underestimated due to cell lysis in high-density culture storage in the medium. Please extend the conclusions. In conclusion, I recommend this manuscript for major revision, due to needed thorough language corrections.

In text comments:

Line 2: Do not divide words in the title

Line 9 Please write what plant model are you using, you mean rice, but you should provide full name, with scientific name and cultivar.

Line 10: Consider rewriting, I could suggest: Screening of Bacillus isolates antagonistic towards….

Please use shorter sentences.

Line 13: There is a dot after Aspergillus.

Line 30 China produces

Line 38: Food cannot be resistant to fungicides, the pathogens can gain such resistance.

Line 45: Please mention the production of spores/ or formulation survival as it is an extremally important feature for biological plant protection

Line 48: Other should not be written in capital, biocontrol mechanisms are not limited to the production of secondary metabolites, please elaborate on the mode of action of biocontrol Bacillus

Line 52: This sentence suggests that the antifungal proteins will be enlisted.

Line 55: Please include more information on the inhibition of pathogenesis it is an interesting mode of action, proportionally understudied, but it does not mean that such studies do not exist. Presenting the published data on other strains and how they suppress the pathogenicity will serve as a good background for your study.

Line 66: When scientific names appear for the first time in the abstract, main text, and in each table or figure caption full name should be used. Please write what media were used for the growth as it affects the phenotype.

Line 72 The word “some” rarely finds its use in scientific publications please e specific, give a number or write the results from biochemical tests presented in table 1.

Line 86: Please describe the V-P assay

Line 87 Please provide a higher resolution version of this figure

Line 88: The current trends indicate that 16S is not enough to assign the strain to species level, at least one more gene should be used, otherwise it allow to assign the strain to genus.

Line 92: Please be more precise, from the photography it is visible that the tested inhibitory effect was of the extract, please specify how the extract was obtained.

Line 94: Please upload subfigures with visible scale bars description or describe the length of the scale bars in the figure caption.

Line 99: Be more specific

Line 105 Describe the lanes in that figure, and give more information how the extracts were prepared and visualized in the figure caption.

Line 113: it is unclear what TK or D stands for.

Line 129: Provide more information on interpreting the figure, otherwise it is redundant.

Line 130: Title unclear

Line 130: Please refer the number of DEGs to the number of genes in that category present in the genome.

Line 145: Please include more information, the figure caption should suffice to interpret the presented figure.

Line 270 bacterial hyphae diameter???

Line 283: Please provide citations for the primers.

Line 304: A. tubingensis is not a bacteria and I am not sure what “bacterial cake” is.

Line 308: The term “plate standoff” is extremely colloquial please restrain yourself from using that and similar phrases in publication.

Line 310: “mycelial crawls”???

Line 313: Please provide producer information for used reagents and keep one format for °C.

Line 459: Please provide more literature regarding the topic of your manuscript, Please expand the introduction and refer your data to similar research adding appropriate citations. 36 citations for such a long article on a topic that is widely studied seem to be very little.

Line 321: “Seed liquid” ???

Line 330 : Use rcf, not rpm’s

Line 332: “sterile fermentation broth” Do you mean sterile spent medium?

Line 355: Please be consistent in the form methods, in other methods, you used the past tense passive voice

Line 376: Change citation style

Line 398: repetition

Line 415 Which figure

Line 417: Please add program distributor information

Line 431: What are the 2 first primers used for

Line 460: Add access date

Figure S1 and S2: Add the description to those figures

Figure S2. Please use xy plot not linear for such data

Figure S3: Use outer inner circle not up or down, as it depends on the part of the graph analyzed. The Down-down ratio seems to be a mistake.

Comments on the Quality of English Language

I would recommend that the abstract and whole manuscript undergo detailed linguistic corrections. There are sometimes very long sentences that should be separated. Please avoid colloquial phrases.

Author Response

(The authors gave the same response as above.)

Round 2

Reviewer 1 Report

Comments and Suggestions for Authors

I thank the authors for their attentive attitude to the reviewer's comments. However, it is necessary to make changes in the "Introduction" section: it is necessary to explain why this particular rice variety was chosen.

Comments on the Quality of English Language

Minor editing of English language required.

Author Response

Dear Reviewer,

Thank you very much for useful suggestions on our manuscript. It is very kind of you to give us the chance to improve our manuscript. We revised the manuscript carefully according to the suggestions. And we highlight some changes by using the red text. Please see the attachment.

We would be grate for any suggestions given by you, and we will consider them about. Thank you!

Kind regards,

Yours Sincerely,

Sun Weihong,Zhao Tianyuan 

College of Agricultural Engineering, Jiangsu University

Zhenjiang, Jiangsu Province,212013

P.R. China

Reviewer 2 Report

Comments and Suggestions for Authors

General information:

I would like to thank the authors for responding to my comments. I believe 16S gene sequencing is insufficient to assign the strain to the species level, but I understand that many publications use only this gene for that purpose even today. Since the assignment of strain to species is not the major object of this manuscript this level of accuracy can be accepted. However, I would recommend using the genus assignment with the strain number and in your future studies plan the sequencing of at least one more housekeeping gene. Thank you for extending the introduction part and addressing my major concerns. I would still recommend extending the figure captions, especially for Figures S1 and S2 as figure captions should be sufficient to interpret the data presented in the figure. The language of the manuscript has been sufficiently improved and I have not noticed serious issues. I understand that you set your centrifugation parameters by setting rpms, not rcf but this should be avoided since for larger rotors the same rpm signifies a higher centrifugation force. Therefore I recommend calculating Rcf for used rotors (which can be done by setting the used centrifuge to a given rpm value and changing to rcf) or at least giving the used rotor parameters. In conclusion, I consider this manuscript acceptable for publication after minor revisions.

In-text comments:

Figures S1 and S2 need more detailed figure captions.

Line 12 paddy should not be in italics

Line 45 genus, not strain

Line 329 dual

Line 366: I you have been using rpm settings on your centrifuge, recalculate it to rcf based on your rotor radius, as the centrifuges may differ in rotor size and the same rpm on different rotors may signify different centrifugation strength. Keep rpm for rotary shakers.

Author Response

(The authors gave the same response as above.)
